# Comparative Analysis of Traumatic Cardiac Arrest: Role of Early Intervention and Care Pathway

**DOI:** 10.3390/healthcare13192532

**Published:** 2025-10-07

**Authors:** Sung Woo Jang, Jae Sik Chung, Pil Young Jung

**Affiliations:** 1Department of Surgery, Soonchunhyang University Seoul Hospital, Seoul 04401, Republic of Korea; longvoiceactor@gmail.com; 2Department of Traumatology, Department of Surgery, Wonju Severance Christian Hospital, Yonsei University Wonju College of Medicine, Wonju 26426, Republic of Korea; gsjaesik@yonsei.ac.kr

**Keywords:** traumatology, cardiac arrest, cardiopulmonary resuscitation, survival analysis

## Abstract

(1) Background: This study aimed to assess the characteristics and factors influencing 72 h survival after traumatic cardiac arrest (TCA), comparing out-of-hospital TCA (oTCA) with in-hospital TCA (iTCA). (2) Methods: This is a retrospective review of 286 patients with TCA admitted to the regional trauma center (RTC) in Gangwon Province, Korea, between 2013 and 2019. (3) Results: Transfer from another hospital (hazard ratio [HR] 0.86 [0.76–0.97]) and longer duration between accident and cardiopulmonary resuscitation (CPR) (HR 0.95 [0.90–0.99]) were associated with lower 72 h mortality. Transfer showed a significant association with lower 72 h mortality in all patients and in the high-injury-severity-score (ISS) group, but not in the low-ISS group. Subgroup analysis indicated that patients transferred from another hospital had significantly lower HR than directly admitted patients to the RTC for oTCA occurrence (HR 0.36 [0.23–0.57]), total CPR duration > 30 min (HR 0.34 [0.23–0.52]), and accident-to-CPR duration < 30 min (HR 0.25 [0.11–0.55]). Additionally, shorter distances from the accident site to the first hospital were associated with lower relative HRs. (4) Conclusions: Considering the extremely poor outcomes of TCA, basic resuscitation and evaluation at nearby medical institutions rather than immediate transfer to specialized trauma centers, particularly when TCA occurs or is anticipated, are important. Early damage-control resuscitation at a nearby hospital can impact on improving the survival rate of patients with TCA.

## 1. Introduction

Trauma is a major cause of death, particularly in younger populations, and represents a significant public health concern as it disproportionately affects socioeconomically active age groups [1]. Traumatic cardiac arrest (TCA) is associated with a low survival rate [2,3,4]. Globally, TCA survival rate was 5.1–7.7% [2,5,6,7]. In the Republic of Korea (Korea), the 24 h TCA survival rate was 1.1% in 1997 and increased to 3.8% in 2020 [4,8]. Despite this improvement over the past two decades, the survival rates have remained comparatively lower than those in other countries.

The most distinctive feature and primary cause of TCA is hypovolemic shock, resulting from blood loss caused by damage to various organs [9,10]. With the improvement in the understanding of the pathophysiological mechanisms underlying TCA, many institutions have adopted damage-control resuscitation, leading to better outcomes in specific patient groups, such as those with penetrating injuries [6,9,10,11]. However, unlike in other countries, in Korea, penetrating injuries account for only 0.7–1.8% of TCAs, and the most frequent cause of traumatic damage is blunt trauma [12,13,14]. Hence, further evaluation is required for blunt trauma injuries and for appropriate patient transfer in the Korean trauma system.

A regional trauma center (RTC) is a dedicated trauma-care facility equipped with the infrastructure, equipment, and personnel to perform emergency surgery immediately upon a patient’s arrival and to deliver optimal care for severe trauma, including cases with multiple fractures and hemorrhage [15]. In Korea, the system began in 2012 with five centers; as of 2025, a total of 17 RTCs have been designated. However, limitations in personnel, resources, and geography make it difficult for all trauma patients to be treated at trauma centers. As a result, many patients are managed at non-trauma centers or are stabilized and then transferred to a trauma center or another facility [16].

The outcomes of cardiac arrest can vary significantly, depending on the underlying mechanisms and affected areas. Most patients with cardiac arrest die within a few hours; however, patients who survive beyond this critical timeframe may have a more favorable long-term neurological prognosis [17,18,19]. Consequently, our study specifically targeted patients who developed cardiac arrest within 24 h of traumatic injury and their 72 h survival, representing a distinct subgroup compared with typical immediate TCA cases.

## 2. Materials and Methods

### 2.1. Regional Characteristics

Gangwon Province in Korea is characterized by over 90% mountainous terrain, which poses challenges in the transfer of traumatically injured patients, compared to other regions in Korea. This study was conducted at the RTC in Gangwon, which functions as a definitive care facility for treating critically injured trauma patients in Gangwon and its surrounding areas (Figure 1).

### 2.2. Study Design and Population

We conducted a single-institution retrospective review of the medical records of patients with TCA who were admitted to Wonju Severance Christian Hospital, an RTC in Gangwon, between May 2013 and December 2019. Among 545 patients with TCA, those aged < 18 years (N = 9), those with TCA caused by hanging, drowning, or poisoning (N = 190), and those with cardiopulmonary resuscitation (CPR) duration of less than 5 min (N = 60) were excluded to reduce bias from early mortality following the accident. Finally, 286 patients were included in this study. Patients were analyzed according to 72 h survival status (survivors vs. non-survivors). The location of occurrence (in-hospital [iTCA] vs. out-of-hospital [oTCA]) was treated as an independent variable in subsequent analyses (Figure 2).

This study was conducted in accordance with the ethical principles of the Declaration of Helsinki and approved by the Institutional Review Board of Yonsei University Wonju Severance Christian Hospital (2021-0025-001). Informed consent was waived due to the study’s retrospective nature.

### 2.3. Data Collection

Patient demographics, including age, sex, inebriation status, personal protective equipment use, mode of transfer (i.e., patients admitted directly to RTC; direct group vs. patients transferred from another hospital; transfer group), injury severity score (ISS), and abbreviated injury score (AIS), were obtained. Data on therapeutic characteristics included the location of occurrence of TCA, transfer location, distance from the accident site to the initial transfer hospital, distance from the initial hospital to the RTC, time from accident to CPR, out-of-hospital CPR (OHCPR) duration, out-of-hospital cardiac arrest (OHCA) duration, and in-hospital cardiac arrest (IHCA) duration, defined as the duration of active resuscitation (IHCPR). Time from accident to discharge was also collected. In cases where patients experienced both out-of-hospital and in-hospital arrests, classification was based on the location of the first arrest. Return of spontaneous circulation (ROSC) was defined as the restoration of spontaneous circulation following the index traumatic cardiac arrest, irrespective of location (prehospital, referring hospital, or our regional trauma center). Arrival time was categorized into two groups: on-duty (weekdays from 08:00 to 17:30) and off-duty (weekdays from 17:30 to 08:00, weekends, or holidays). Hospitalization event data included spontaneous circulation recovery, surgical procedures, transfusions, and mortality.

### 2.4. Statistical Analysis

Continuous variables are expressed as mean ± standard deviation for normally distributed variables and as median with interquartile range for non-normally distributed variables, and categorical variables as frequencies and percentages. Continuous data were tested for normal distribution using the Shapiro–Wilk test and compared using Student’s *t*-test or Mann–Whitney U test, as appropriate. The chi-squared test was used for categorical variables. Kaplan–Meier analysis was used to estimate event-free survival. Cox proportional hazards regression analysis was performed. Variables were selected based on clinical relevance (age, sex, ISS, CPR duration, duty time) and prior literature. Three models were constructed with incremental adjustment: Model 1 (baseline characteristics), Model 2 (baseline + pre-hospital factors), and Model 3 (fully adjusted, including in-hospital variables). We conducted a subgroup Cox proportional analysis to compare the direct group (patients directly transferred to the RTC) and the transfer group (those transferred from another hospital). The non-linear relationships between hazard ratio (HR) and transfer duration from the accident site to the RTC, distance from the accident site to the RTC, and distance from the accident site to the initial transfer hospital were evaluated using restricted cubic spline curve analysis. A *p*-value < 0.05 was considered statistically significant. Statistical analyses were performed using R statistical software (version 4.3.1; R Foundation for Statistical Computing, Vienna, Austria).

## 3. Results

### 3.1. Patient Characteristics

Most cases (89.9%) occurred in the south-west area, and over half of the cases (55.2%) occurred within 25 km of the RTC. The number of transfer patients decreased as the distance from the RTC increased. Car accidents were the most common cause of trauma (50.7%), and hemothorax was the leading cause of death (29.0%). Among the patients, 127 had iTCA, with 19 surviving for 72 h and 11 achieving complete survival. Additionally, 159 patients had oTCA, with 13 surviving for 72 h and 7 achieving complete survival.

The patients were categorized into two groups based on their outcomes: those who survived for more than 72 h (72 h survival) and those who died within 72 h (72 h death). There was a significant difference in ISS between the groups, but no significant differences were observed in AIS1 (head and neck), AIS3 (thorax), AIS4 (abdomen and pelvis), or AIS5 (extremities). TCA occurred at other hospitals in four patients (1.4%), in the emergency department in 81 patients (28.4%), and after admission in 42 patients (14.7%). Among all patients, 179 (62.6%) were directly transferred to the RTC, whereas 107 (37.4%) were transferred to the RTC via other hospitals (Table 1). The total collapse, total CPR, OHCPR, and IHCA durations were significantly shorter in the 72 h survival group; however, OHCA duration, time to CPR, time to RTC, distance to RTC, distance to the initial transfer hospital, and distance from the initial transfer hospital to RTC were not significantly different between the two groups (Table 2).

### 3.2. Cox Proportional Hazards Regression Analysis

The transfer group exhibited a significant association with lower 72 h mortality in all models compared to the direct group; in the fully adjusted model (Model 3), HR was 0.86 (95% CI 0.76–0.97, *p* = 0.014). The HR for iTCA was significant compared to that for oTCA in Models 1 and 2, but not in Model 3 (HR 1.02, 95% CI 0.91–1.14, *p* = 0.714). Longer accident-to-CPR time was also associated with lower 72 h mortality (HR 0.95, 95% CI 0.90–0.99, *p* = 0.050). Additionally, the arrival of patients during on-duty hours was associated with lower 72 h mortality (HR 0.73, 95% CI 0.54–0.97, *p* = 0.028); however, this effect was only significant in Model 3 (Table 3).

### 3.3. Kaplan–Meier Survival Analysis

The survival probability of the high-ISS group (≥26) was significantly lower in all patients, as well as in the iTCA and oTCA subgroups. The transfer group showed a significantly higher survival probability than the direct group for all patients. This finding was consistent in the high-ISS subgroup; however, there was no statistically significant difference in survival in the low-ISS subgroup (Figure 3).

### 3.4. Subgroup Analysis Based on Transfer Status

The transfer and direct groups were analyzed to investigate any interactions (Figure 4). In the fully adjusted model, transfer from another hospital was associated with a lower hazard of death within 72 h across most subgroups, with hazard ratios below 1 for nearly all categories (e.g., sex, age, duty time, and first-hospital distance). The magnitude of this protective association varied by subgroup—it was stronger in patients with oTCA, accident-to-CPR duration < 30 min, and total CPR duration > 30 min, and was attenuated when accident-to-CPR exceeded 30 min, yet remained protective. Notably, patients with ISS ≥ 30 also demonstrated a survival benefit with transfer.

### 3.5. Restricted Cubic Spline Curve Analysis for Distance

Using restricted cubic spline analysis based on the prediction technique, the association between continuous variables and mortality rates, as well as the determined cut-off points, was investigated (Figure 5). The HR of the distance from the accident site to the initial transfer hospital, including the RTC, showed an inverted U-shaped pattern for all patients, with cut-offs at 5.1 km and 50.9 km. More specifically, for distances between 5.1 km and 50.9 km, the HR was greater. As the distance from the accident site to the initial transfer hospital increased, the relative HR for the oTCA group exhibited a generally linear increase, with a cut-off at 5.6 km. Analysis of the distance from the accident site to the RTC and the mortality rate in the entire patient population revealed an interesting finding: the relative HR decreased linearly with increasing distance. This consistent pattern was also observed when focusing on the oTCA group, with cut-offs at 31.9 km for all patients and 23.2 km for the oTCA group.

## 4. Discussion

TCA is a life-threatening condition, and its underlying pathophysiological mechanisms differ from those of medical cardiac arrest (MCA). The cause of TCA must be determined simultaneously with patient resuscitation. Most causes of MCA are cardiogenic, but TCA is caused by factors outside the heart, such as hypovolemia (from blood loss), hypoxia (from respiratory system damage, including the oropharynx), physical obstruction of cardiac output, and central nervous system damage [9]. A prerequisite of TCA is relatively normal heart function before the accident, making TCA a hyperacute condition. Because many TCAs are caused by extracardiac factors, a “golden period” exists during which the damage can be reversed, unless there is a serious central nervous system condition. In hypovolemic situations, which are common among patients with TCA, maintaining adequate cardiac output during chest compressions becomes challenging, resulting in less effective CPR than that in patients with MCA [20].

The survival rate of patients with TCA caused by blunt or penetrating injuries remains debatable. Several studies have reported no significant differences in survival rates between groups [20,21,22]. Nevertheless, in recent years, research on TCA pathophysiology and damage-control resuscitation has made significant progress, leading to mounting evidence that the survival rate of patients with TCA after a penetrating injury is higher [9,23]. This could be a reason for the low TCA survival rate in Korea, as the occurrence of penetrating injuries is extremely rare (<2%). For this reason, we determined that subgroup analysis by trauma mechanism was not feasible; therefore, we focused specifically on blunt trauma TCA, which reflects the epidemiological reality in Korea.

This subgroup differs from many prior TCA studies, which primarily analyzed cardiac arrest occurring immediately post-injury. Our findings should be interpreted within this narrower clinical context. No associations were found in the multivariate Cox proportional hazards regression analysis for 72 h survival between iTCA and oTCA, contradicting previous results stating that oTCA had a significant effect on mortality [24,25,26]. To compensate for the high mortality rate in arrests occurring immediately after the accident, we explicitly excluded patients with CPR < 5 min. Therefore, this result may be due to the longer total CPR time, higher ISS, and exclusion of patients with early arrest.

The discrepancy between univariate comparisons and Cox regression findings for pathway likely reflects the different statistical properties of the methods. Cox regression accounts for time-to-event data and censoring, while univariate analyses do not. Residual confounding cannot be ruled out, and pathway should be interpreted with caution. In the transfer group, the Kaplan–Meier survival analysis showed a significantly high survival probability. However, in the low-ISS subgroup, there was no significant difference between the direct and transfer groups. Additionally, subgroup analyses indicate that transfer is broadly protective across most strata, with hazard ratios below 1 observed in nearly all categories, including sex, age, duty time, and first-hospital distance. While the association was relatively stronger in patients with oTCA, shorter accident-to-CPR intervals (<30 min), and prolonged total CPR (>30 min), the effect remained protective even when accident-to-CPR exceeded 30 min. Importantly, a survival benefit was also evident among patients with ISS ≥30, which aligns with the overall consistency of the transfer effect across subgroups. Thus, basic life support is necessary for such patients in nearby hospitals. In the restricted cubic spline curve analysis, a significant association with lower 72 h mortality was observed for shorter distances to the initial transfer hospital, whereas a similar significant association with lower 72 h mortality was observed for longer distances to the RTC. As the distance from the accident site to the RTC increased, the likelihood of transfer to other hospitals also increased. Patients who died at the transfer location were excluded from this study, and those closer to the RTC, selected as the first transfer hospital, experienced more deaths. Consequently, this created a selection bias, with patients farther away from the RTC showing a higher survival rate. Nevertheless, a significant association with lower 72 h mortality was associated with shorter distances to the initial transfer hospital, which was stronger in the oTCA group. This highlights the importance of rapid basic life support in patients with severe injuries.

The direct transfer of patients with severe trauma to specialized trauma centers without passing through a nearby medical institution reduces the incidence of preventable death [13,27,28]. However, it is important to note that the population in this study comprised patients who experienced cardiac arrest within 24 h. Consequently, a different trend was observed, with a lower 72 h mortality rate when the initial treatment and evaluation were conducted at a nearby medical facility. Considering the high mortality rate of TCA, this finding indicates that basic resuscitation should take precedence in the field or at a nearby medical facility rather than extensive treatment at a specialized trauma center, especially in severely injured patients with a high likelihood of TCA.

This study was retrospective and single institution, which may introduce selection bias. In particular, patients stable enough for transfer may have inherently had a better prognosis, potentially overestimating the benefit of transfer. In addition, the number of 72 h survivors was small (N = 32), raising concerns about overfitting in multivariable Cox regression. So, these findings must therefore be interpreted with caution. To mitigate these limitations, we reduced the number of covariates, performed sensitivity analyses, and acknowledge this as a limitation. And especially to address selection bias and enhance the overall validity of the findings, diverse subgroup analyses were conducted. However, to obtain more accurate results, future research should involve multiple institutions from regions participating in TCA studies. This multi-institutional approach will provide a broader perspective and increase the generalizability of the results.

## 5. Conclusions

The TCA mortality rate is lower when patients are initially transferred to a nearby hospital and subsequently transferred to the RTC. Considering the higher mortality rate in patients with TCA and situations where TCA has occurred or is strongly anticipated, it is strongly recommended to prioritize primary damage-control resuscitation at a nearby medical institution rather than transferring the patient to a specialized trauma center. This approach can lead to improved survival outcomes and better management of patients with critical injuries.

## Figures and Tables

**Figure 1 healthcare-13-02532-f001:**
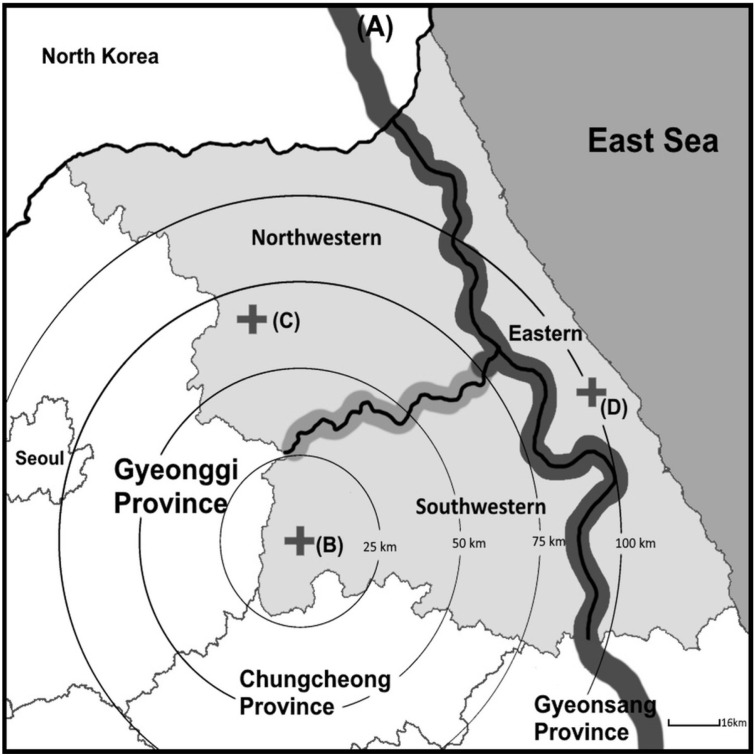
Map of Gangwon Province. (**A**) Main mountain range of Korea. The average height is 1248 m, and it is a boundary dividing eastern Gangwon from western Gangwon. (**B**) Regional trauma center in Gangwon. (**C**,**D**) Regional medical centers in northwestern and eastern regions.

**Figure 2 healthcare-13-02532-f002:**
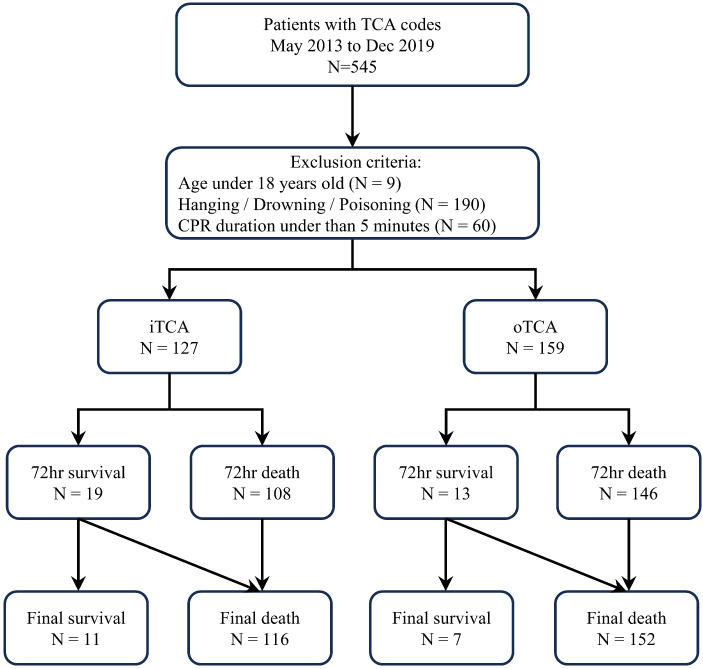
Flowchart of this study. TCA, traumatic cardiac arrest; CPR, cardiopulmonary resuscitation; iTCA, in-hospital TCA; oTCA, out-of-hospital TCA.

**Figure 3 healthcare-13-02532-f003:**
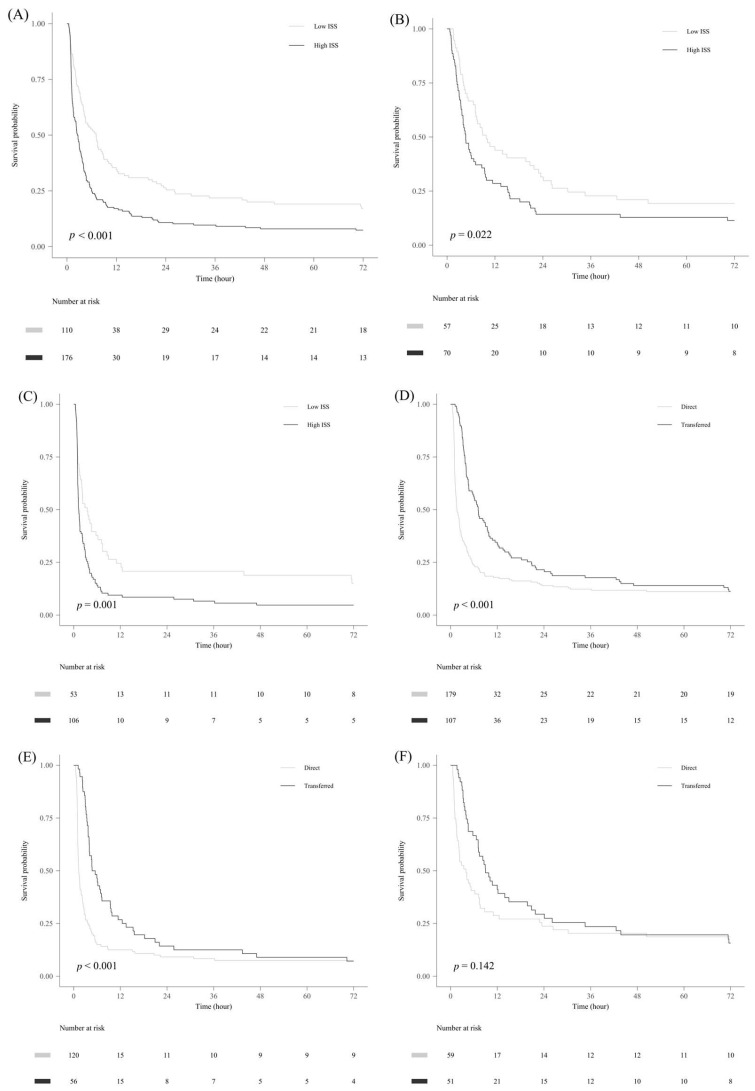
Kaplan–Meier survival analysis. (**A**) Comparison between low ISS (<26) and high ISS (≥26) in all patients with TCA. (**B**) Comparison between low ISS and high ISS in all patients with iTCA. (**C**) Comparison between low ISS and high ISS in all patients with oTCA. (**D**) Comparison between the direct group (directly transported to the RTC) and the transferred group (transferred from another hospital) in all patients with TCA. (**E**) Comparison between direct group and transferred group in all patients with TCA with high ISS. (**F**) Comparison between the direct and transferred groups in all patients with TCA with low ISS. ISS, injury severity score; TCA, traumatic cardiac arrest; oTCA, out-of-hospital TCA; iTCA, in-hospital TCA.

**Figure 4 healthcare-13-02532-f004:**
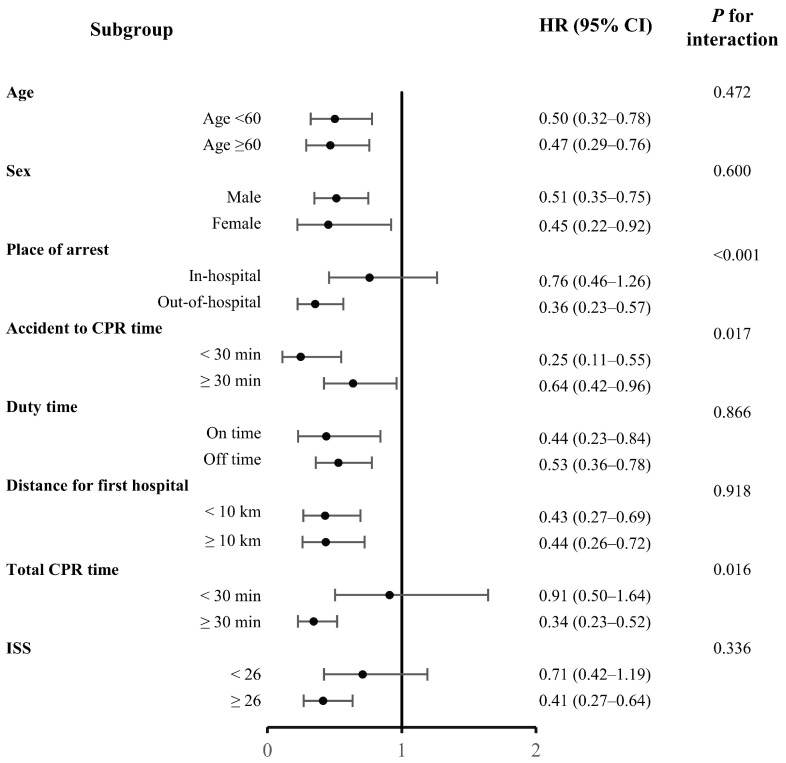
Subgroup analysis for pathway (transfer vs. direct)—fully adjusted Cox model. Hazard ratios (squares) and 95% confidence intervals (lines) are shown for each subgroup; the vertical line indicates HR = 1. HRs < 1 favor transfer. The transfer effect was broadly consistent across most strata, with relative differences in effect size noted (see Section 3.4).

**Figure 5 healthcare-13-02532-f005:**
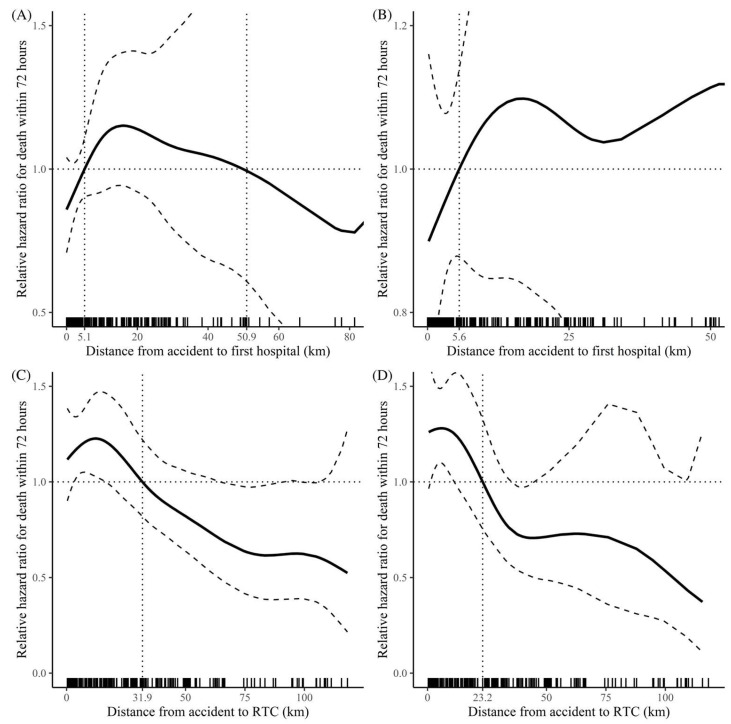
Restricted cubic spline curve analysis for relative hazard ratio for death within 72 h. (**A**) The distance from the accident site to the first hospital in all patients. (**B**) Distance from the accident site to the first hospital in the oTCA group. (**C**) Distance from the accident site to the RTC in all patients. (**D**) Distance from the accident site to the RTC in the oTCA group. oTCA, out-of-hospital traumatic cardiac arrest; RTC, regional trauma center.

**Table 1 healthcare-13-02532-t001:** Demographic characteristics of included patients.

	All Patients (N = 286)	72 h Survival (N = 32)	72 h Death (N = 254)	*p* Value
Sex				0.970
Male	193 (67.5%)	21 (65.6%)	172 (67.7%)	
Female	93 (32.5%)	11 (34.4%)	82 (32.3%)	
Age (years)	59.0 [45.0; 71.0]	59.0 [45.5; 65.0]	59.0 [45.0; 73.0]	0.408
Ways of visit				0.443
EMS	179 (62.6%)	21 (65.6%)	158 (62.2%)	
Hospital ambulance	20 (7.0%)	4 (12.5%)	16 (6.3%)	
Private ambulance	55 (19.2%)	6 (18.8%)	49 (19.3%)	
Air transportation	30 (10.5%)	1 (3.1%)	29 (11.4%)	
Private vehicle	2 (0.7%)	0 (0.0%)	2 (0.8%)	
Drunken state				0.385
Yes	14 (4.9%)	3 (9.4%)	11 (4.3%)	
No	164 (57.3%)	19 (59.4%)	145 (57.1%)	
Unknown	108 (37.8%)	10 (31.2%)	98 (38.6%)	
Protection				0.298
Yes	37 (12.9%)	5 (15.6%)	32 (12.6%)	
No	182 (63.6%)	23 (71.9%)	159 (62.6%)	
Unknown	67 (23.4%)	4 (12.5%)	63 (24.8%)	
ISS	34.0 [22.0; 75.0]	24.5 [17.0; 32.0]	34.0 [24.0; 75.0]	<0.001
AIS1 (head and neck)	2.0 [0.0; 4.0]	2.0 [0.0; 4.0]	2.0 [0.0; 4.0]	0.851
AIS2 (face)	0.0 [0.0; 0.0]	0.0 [0.0; 2.0]	0.0 [0.0; 0.0]	0.006
AIS3 (thorax)	3.0 [0.0; 4.0]	3.0 [0.0; 4.0]	3.0 [0.0; 4.0]	0.353
AIS4 (abdomen and pelvis)	0.0 [0.0; 2.0]	0.0 [0.0; 2.0]	0.0 [0.0; 2.0]	0.379
AIS5 (extremities)	1.0 [0.0; 3.0]	0.0 [0.0; 2.0]	1.0 [0.0; 3.0]	0.329
AIS6 (external)	0.0 [0.0; 0.0]	0.0 [0.0; 0.0]	0.0 [0.0; 0.0]	0.022
Place of cardiac arrest				0.063
Out of hospital	159 (55.8%)	13 (40.6%)	146 (57.7%)	
First hospital	4 (1.4%)	1 (3.1%)	3 (1.2%)	
ED of RTC	81 (28.4%)	15 (46.9%)	66 (26.1%)	
Ward of RTC	42 (14.7%)	3 (9.4%)	39 (15.4%)	
Pathway				1.000
Directly transported	179 (62.6%)	20 (62.5%)	159 (62.6%)	
Transferred	107 (37.4%)	12 (37.5%)	95 (37.4%)	
Duty time				1.000
Off-duty time	206 (72.0%)	23 (71.9%)	183 (72.0%)	
On-duty time	80 (28.0%)	9 (28.1%)	71 (28.0%)	

EMS, emergency medical service; ISS, injury severity score; AIS, abbreviated injury scale; ED, emergency department.

**Table 2 healthcare-13-02532-t002:** Therapeutic characteristics of included patients.

	All Patients (N = 286)	72 h Survival (N = 32)	72 h Death (N = 254)	*p* Value
Total collapse time (min)	37.5 [22.0; 55.0]	12.0 [5.0; 25.5]	40.0 [28.0; 57.0]	<0.001
Total CPR time (min)	35.0 [21.0; 49.0]	10.0 [5.0; 21.0]	36.0 [25.0; 51.0]	<0.001
OHCA time (min) (N = 159)	24.0 [14.0; 37.0]	17.0 [12.0; 29.0]	24.0 [15.0; 39.0]	0.123
OHCPR time (min) (N = 159)	17.0 [10.0; 26.5]	14.0 [7.0; 19.0]	17.0 [10.0; 27.0]	0.003
IHCA time (min)	24.0 [12.0; 32.0]	6.0 [2.5; 10.5]	27.0 [16.0; 34.0]	<0.001
Time to CPR time (min)	36.0 [14.0; 130.0]	33.5 [13.5; 170.5]	36.5 [14.0; 127.0]	0.974
Distance to RTC (km)	18.1 [3.5; 42.7]	9.6 [2.3; 45.0]	19.5 [3.9; 42.7]	0.368
Distance to 1st hospital (km)	4.5 [1.8; 18.0]	2.6 [1.6; 7.7]	5.5 [1.8; 19.0]	0.145
Distance from 1st hospital to RTC (km) (N = 107)	41.1 [32.8; 61.3]	41.1 [32.8; 88.4]	41.1 [32.8; 49.4]	0.361
ROSC	98 (34.3%)	32 (100.0%)	66 (26.0%)	<0.001
Operation	73 (25.5%)	19 (59.4%)	54 (21.3%)	<0.001
Chest tube	126 (44.1%)	16 (50.0%)	110 (43.3%)	0.596
Pelvic binder	28 (9.8%)	1 (3.1%)	27 (10.6%)	0.303
ECMO	5 (1.7%)	2 (6.2%)	3 (1.2%)	0.178
REBOA	12 (4.2%)	0 (0.0%)	12 (4.7%)	0.430
Angiography	10 (3.5%)	2 (6.2%)	8 (3.1%)	0.697
Transfusion (RBC) (mL)	3200.0 [0.0; 8400.0]	7720.0 [4020.0; 20,360.0]	3120.0 [0.0; 7520.0]	<0.001
Transfusion (FFP) (mL)	1600.0 [0.0; 6400.0]	4920.0 [2000.0; 19,800.0]	0.0 [0.0; 6400.0]	<0.001
Transfusion (PC) (mL)	0.0 [0.0; 0.0]	4320.0 [0.0; 7800.0]	0.0 [0.0; 0.0]	<0.001
Place of death				<0.001
Emergency department	168 (58.7%)	0 (0.0%)	168 (66.1%)	
Intensive care unit	81 (28.3%)	0 (0.0%)	70 (27.5%)	
Operation room	12 (4.2%)	0 (0.0%)	12 (4.7%)	
DAMA	5 (1.7%)	2 (6.2%)	3 (1.2%)	
General ward	1 (0.3%)	0 (0.0%)	1 (0.4%)	
Organ harvest	1 (0.3%)	1 (3.1%)	0 (0.0%)	

CPR, cardiopulmonary resuscitation; OHCA, out-of-hospital cardiac arrest; OHCPR, out-of-hospital cardiopulmonary resuscitation; IHCA, in-hospital cardiac arrest; RTC; regional trauma center; ROSC, return of spontaneous circulation; ECMO, extracorporeal membrane oxygenation; REBOA, resuscitative endovascular balloon occlusion of the aorta; RBC, red blood cell; FFP, fresh frozen plasma; PC, platelet concentration; DAMA, discharge against medical advice.

**Table 3 healthcare-13-02532-t003:** Cox proportional hazards regression analysis for entire patient cohort.

Variable	Unadjusted		Model 1		Model 2		Model 3	
	HR (95% CI)	*p*	HR (95% CI)	*p*	HR (95% CI)	*p*	HR (95% CI)	*p*
Age	1.00 (0.99–1.00)	0.182	1.00 (0.99–1.01)	0.826	1.00 (1.00–1.01)	0.320	1.01 (1.00–1.01)	0.105
Sex								
Female	(Reference)		(Reference)		(Reference)		(Reference)	
Male	1.07 (0.83–1.40)	0.593	0.93 (0.71–1.22)	0.606	0.93 (0.71–1.22)	0.605	0.98 (0.75–1.28)	0.881
Pathway								
Direct	(Reference)		(Reference)		(Reference)		(Reference)	
Transferred	0.78 (0.70–0.86)	<0.001	0.78 (0.71–0.86)	<0.001	0.86 (0.77–0.96)	0.006	0.86 (0.76–0.97)	0.014
Place of arrest								
In	(Reference)		(Reference)		(Reference)		(Reference)	
Out	1.16 (1.08–1.25)	<0.001	1.16 (1.08–1.25)	<0.001	1.12 (1.04–1.21)	0.003	1.02 (0.91–1.14)	0.716
Accident to CPR time	0.88 (0.84–0.94)	<0.001			0.93 (0.88–0.98)	0.007	0.95 (0.90–0.99)	0.050
Duty time								
Off-duty time	(Reference)				(Reference)		(Reference)	
On-duty time	0.85 (0.65–1.12)	0.250			0.83 (0.62–1.09)	0.182	0.73 (0.54–0.97)	0.028
Distance to 1st hospital	1.00 (0.99–1.00)	0.704			1.00 (0.99–1.00)	0.518	1.00 (0.99–1.00)	0.607
Total CPR time	1.00 (1.00–1.01)	0.057					1.00 (1.00–1.01)	0.018
ISS	1.03 (1.02–1.04)	<0.001					1.03 (1.02–1.03)	<0.001

HR, hazard ratio; CI, confidence interval; In, In-hospital arrest; Out, Out-of-hospital arrest; CPR, cardiopulmonary resuscitation; ISS, injury severity score. Model 1: adjusted age, sex, pathway, place of arrest. Model 2: Model 1 + accident to CPR time, duty time, distance for first hospital. Model 3: Model 2 + total CPR time, ISS.

## Data Availability

The data presented in this study are available on request from the corresponding author. The data are not publicly available due to privacy of enrolled patients.

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
