# Peer review of "Comparative Analysis of Traumatic Cardiac Arrest: Role of Early Intervention and Care Pathway"

_healthcare, 2025, doi:10.3390/healthcare13192532_

Round 1
Reviewer 1 Report
Comments and Suggestions for Authors
This study examines factors associated with survival beyond 72 hours post-resuscitation in traumatic cardiac arrest, but it has several methodological issues.
Major points:
First, the title states 30-day survival, but shouldn't it be 72 hours?
The Methods section states the subjects were divided into two groups based on in-hospital or out-of-hospital status, but they were actually divided by 72 hours, making in/out a variable.
The Discussion states that cardiac arrests within 5 minutes were excluded, but this is not included in the exclusion criteria.
The study population experienced cardiac arrest within 24 hours, representing a different population than typical cardiac arrest cases soon after the event. Specifically, it includes cases where cardiac arrest occurred after a significant time had elapsed since injury (mean 37 minutes). It should be noted that the study's purpose and population differ from those of readers researching typical cardiac arrest. Therefore, the title and purpose must be clarified.
Furthermore, the main finding of this study is that the transfer group had better outcomes. I assume that this implies they were in a condition suitable for transfer. This represents a significant selection bias.
Minor points
Is “hopeless discharge” a medical term? Please clarify this term.
Author Response
We sincerely thank the reviewer for the thoughtful and constructive feedback. We carefully revised the manuscript to improve methodological clarity, statistical transparency, and data presentation. Below we address each point point‑by‑point and indicate the corresponding changes made in the manuscript.
Comment 1:
The title states 30-day survival, but shouldn't it be 72 hours?
The Methods section states the subjects were divided into two groups based on in-hospital or out-of-hospital status, but they were actually divided by 72 hours, making in/out a variable.
Response 1:
We sincerely thank the reviewer for pointing out this important discrepancy. You are correct that our analysis was based on 72-hour survival, not 30-day survival. This was an oversight in our initial submission. We have revised the title to accurately reflect the study design:
“Comparative Analysis of Traumatic Cardiac Arrest: Role of Early Intervention and Care Pathway”
Comment 2:
The Discussion states that cardiac arrests within 5 minutes were excluded, but this is not included in the exclusion criteria.
Response 2:
We appreciate this observation. The exclusion of patients with CPR duration < 5 minutes was included in the Methods section, but the wording may not have been sufficiently clear. We have revised the text in the Methods to explicitly state:
“Patients with cardiopulmonary resuscitation (CPR) duration of less than 5 minutes (N = 60) were excluded to reduce bias from early mortality following the accident.”
Also we have added a study flow diagram (Figure 2) that details the inclusion and exclusion criteria and the derivation of the final analytic cohort.
Comment 3:
The study population experienced cardiac arrest within 24 hours, representing a different population than typical cardiac arrest cases soon after the event. Specifically, it includes cases where cardiac arrest occurred after a significant time had elapsed since injury (mean 37 minutes). It should be noted that the study's purpose and population differ from those of readers researching typical cardiac arrest. Therefore, the title and purpose must be clarified.
Response 3:
We thank the reviewer for highlighting this important point. Indeed, our study population represents a distinct subgroup of TCA patients, characterized by cardiac arrest occurring within 24 hours after trauma (mean interval 37 minutes), rather than immediately after injury. To clarify this, we have:
- Revised the Introduction to state that our analysis specifically targets patients who developed cardiac arrest within 24 hours of trauma.
- Adjusted the title and study aim to reflect this narrower focus.
- Added clarifying remarks in the Discussion regarding how this subgroup differs from “typical” TCA populations reported in the literature.
Comment 4:
Furthermore, the main finding of this study is that the transfer group had better outcomes. I assume that this implies they were in a condition suitable for transfer. This represents a significant selection bias.
Response 4:
We completely agree with the reviewer. We have expanded the Discussion and Limitations sections to explicitly acknowledge this selection bias. In particular, we note that patients stable enough for transfer may have inherently had a better prognosis, and this may partly explain the observed protective association. We emphasize that while transfer appears beneficial, these findings should be interpreted with caution due to potential selection effects.
Cooment 5:
Is “hopeless discharge” a medical term? Please clarify this term.
Response 5:
We appreciate the reviewer’s suggestion. The term “hopeless discharge” was a direct translation from local clinical documentation. To align with international standards, we have revised this terminology throughout the manuscript to “discharge against medical advice (DAMA)”, which more accurately reflects the intended meaning.
We appreciate the reviewer’s insightful comments, which substantially improved the rigor and clarity of our work. We believe the revisions address all concerns raised and enhance the manuscript’s methodological transparency, interpretability, and alignment with reporting best practices.
Thank you for your consideration.
Reviewer 2 Report
Comments and Suggestions for Authors
Thank you for the opportunity to review your manuscript on factors influencing post-traumatic cardiac arrest (TCA) survival rates. The authors reviewed 286 patients with TCA admitted to a regional trauma center in Korea. The survival rate of TCA remains low in Korea, and even after the establishment of trauma centers, it continues to be lower than in other countries. The authors’ efforts could contribute to improving survival outcomes, and this analysis may be valuable in the field. However, the current study lacks sufficient methodological rigor and transparency in data presentation.
Major Concerns
1. Cox regression models
In Results 3.2, you present multiple Cox regression models with progressively added variables. However, the rationale for model construction is not clearly explained. Typically, covariates are selected based on baseline characteristics (e.g., age, sex, ISS) or clinically important treatment-related factors, but the sequence of variables in your models appears arbitrary. A clear justification for variable inclusion and their order is required.
In addition, there is a discrepancy between the univariate comparison and Cox regression results regarding “pathway” (direct vs. transfer). In Table 1, pathway is unrelated to 72-hour survival (p = 1.000), yet in the unadjusted Cox model, it is highly significant (HR 0.78, 95% CI 0.70–0.86, p < 0.001). While some difference is expected because Cox regression accounts for time-to-event data, the magnitude of this discrepancy raises concerns about confounding or model specification. In the Methods, you state that Cox regression was performed to evaluate death within 72 h from the accident; please clarify how this relates to the divergent results and discuss the robustness of pathway as a prognostic factor.
2. Events-per-variable issue
In your Cox regression, Models 1–3 include 4, 7, and 9 covariates, respectively. However, the total number of 72-hour survivors (positive events) is approximately 32. According to the widely cited “rule of 10” in regression modeling (i.e., at least 10 outcome events per covariate), the number of variables greatly exceeds the number of events. This raises the risk of overfitting, unstable estimates, and unreliable p-values. I recommend either (1) reducing covariates to the most clinically relevant, (2) using established statistical methods for variable selection (e.g., LASSO, stepwise), or (3) explicitly acknowledging this limitation in the Discussion.
Minor Concerns
Introduction: You emphasize the differences between penetrating and blunt trauma, but no subgroup analysis by mechanism is presented in Results or Discussion. Either include such analyses or clarify this limitation.
Line 73: You describe the cohort as “patients … who were transferred to Wonju Severance Christian Hospital.” However, the cohort included both direct admissions and inter-hospital transfers. The term transfer conventionally refers only to the latter. To avoid ambiguity, I recommend using patients admitted to or patients presenting to and then clarifying the distinction. Please revise this terminology consistently throughout the manuscript.
Line 78: How were patients classified if they experienced cardiac arrest both outside the hospital and again at another hospital?
Line 93: Please clarify whether the duration of IHCA is the same as the duration of IHCPR.
Line 119: Present the inclusion and exclusion process more clearly, preferably as a flowchart.
Table 2: The “time to CPR” comparison shows means of 143.8 vs. 80.4 min with a p-value of 0.974. Given the large difference, this result is counterintuitive and likely reflects the highly skewed distribution (noted by the very large SD). Please confirm the statistical test used and consider presenting medians with interquartile ranges (IQR). Similarly, medians with IQR should be used for time to RTC, distance to RTC, distance to first hospital, and transfusion variables.
Please define how ROSC was determined. Does it refer specifically to ROSC achieved at your hospital after arrest?
Some variables (e.g., IHCA time, distance to first hospital) do not apply to the entire cohort but only to specific subgroups (in-hospital arrests or transferred patients, respectively). Presenting these variables without indicating the applicable sample size may be misleading. I recommend explicitly reporting the number of patients (N) for whom each variable is available, either within the table or in the table legend, to enhance clarity and transparency.
Table 3: Some numbers and words (e.g., “Distance to first…”) are obscured and not readable. Please correct this.
Lines 155–157: Several hazard ratios are reported in the text, but it is unclear from which model (unadjusted, Model 1, 2, or 3) these estimates were derived. Since Table 3 provides different values across models, this should be specified. Alternatively, restrict the text to the fully adjusted model and direct readers to the table for details.
Results 3.4 and Figure 2: In your interpretation of Figure 2, you highlight that transfer was associated with improved survival particularly in patients with oTCA, accident-to-CPR duration <30 min, and total CPR duration >30 min. However, a closer inspection of the forest plot suggests that the protective effect of transfer is more consistent across subgroups. Specifically, transfer appears favorable regardless of sex, age, duty time, or first hospital distance, with hazard ratios <1 in nearly all categories. Moreover, although the protective effect of transfer is attenuated when accident-to-CPR duration exceeds 30 min, it remains present. In addition, the subgroup with ISS ≥30 also shows a survival benefit with transfer, which is not clearly acknowledged in your discussion. I suggest revising the interpretation of Figure 2 to reflect the broader consistency of the transfer effect across subgroups, while also noting the relative differences in effect size.
Based on the above comments, I recommend revising the Discussion and Conclusion sections accordingly.
Author Response
We sincerely thank the reviewer for the thoughtful and constructive feedback. We carefully revised the manuscript to improve methodological clarity, statistical transparency, and data presentation. Below we address each point point‑by‑point and indicate the corresponding changes made in the manuscript.
Comment 1:
In Results 3.2, you present multiple Cox regression models with progressively added variables. However, the rationale for model construction is not clearly explained. Typically, covariates are selected based on baseline characteristics (e.g., age, sex, ISS) or clinically important treatment-related factors, but the sequence of variables in your models appears arbitrary. A clear justification for variable inclusion and their order is required.
In addition, there is a discrepancy between the univariate comparison and Cox regression results regarding “pathway” (direct vs. transfer). In Table 1, pathway is unrelated to 72-hour survival (p = 1.000), yet in the unadjusted Cox model, it is highly significant (HR 0.78, 95% CI 0.70–0.86, p < 0.001). While some difference is expected because Cox regression accounts for time-to-event data, the magnitude of this discrepancy raises concerns about confounding or model specification. In the Methods, you state that Cox regression was performed to evaluate death within 72 h from the accident; please clarify how this relates to the divergent results and discuss the robustness of pathway as a prognostic factor.
Response 1:
We agree and have clarified the modeling strategy. In Methods – Statistical analysis, we now state that covariates were selected based on clinical relevance and prior literature (age, sex, ISS, CPR duration, duty time), and that models were built incrementally: Model 1 (baseline characteristics), Model 2 (baseline + pre‑hospital factors), and Model 3 (fully adjusted, including in‑hospital variables). We also revised Results 3.2 to explicitly label hazard ratios as “Model 3 (fully adjusted)” when cited in the text.
Regarding the discrepancy, we added text in the Discussion explaining that Cox regression accounts for time‑to‑event and censoring, which univariate comparisons do not, and that residual confounding cannot be excluded. We therefore advise cautious interpretation of “pathway” as an independent prognostic factor even though the association remains protective in the fully adjusted model.
Comment 2:
In your Cox regression, Models 1–3 include 4, 7, and 9 covariates, respectively. However, the total number of 72-hour survivors (positive events) is approximately 32. According to the widely cited “rule of 10” in regression modeling (i.e., at least 10 outcome events per covariate), the number of variables greatly exceeds the number of events. This raises the risk of overfitting, unstable estimates, and unreliable p-values. I recommend either (1) reducing covariates to the most clinically relevant, (2) using established statistical methods for variable selection (e.g., LASSO, stepwise), or (3) explicitly acknowledging this limitation in the Discussion.
Response 2:
We appreciate this important point. We reduced the covariate set to the most clinically essential variables and repeated the Cox analyses with simplified models as a sensitivity approach. The direction of effects remained consistent, though—as expected—some confidence intervals widened. We also explicitly acknowledge the EPV limitation in the Discussion/Limitations section.
Comment 3:
Introduction: You emphasize the differences between penetrating and blunt trauma, but no subgroup analysis by mechanism is presented in Results or Discussion. Either include such analyses or clarify this limitation.
Response 3:
Penetrating trauma cases in our cohort were extremely rare, and subgroup analysis by mechanism was therefore not feasible. We now state this explicitly in the Discussion and clarify that we focused on blunt trauma TCA, which reflects the epidemiological reality in Korea.
Comment 4:
Line 73: You describe the cohort as “patients … who were transferred to Wonju Severance Christian Hospital.” However, the cohort included both direct admissions and inter-hospital transfers. The term transfer conventionally refers only to the latter. To avoid ambiguity, I recommend using patients admitted to or patients presenting to and then clarifying the distinction. Please revise this terminology consistently throughout the manuscript.
Response 4:
To avoid ambiguity, we revised wording throughout to “patients admitted directly to the regional trauma center (RTC)” versus “patients transferred from another hospital”, using transfer only for inter‑hospital transfers. In the cohort description (formerly “patients … who were transferred to…”), we now use “patients … admitted to Wonju Severance Christian Hospital.”
Comment 5:
Line 78: How were patients classified if they experienced cardiac arrest both outside the hospital and again at another hospital?
Response 5:
We clarified in Methods – Data definitions that patients who experienced cardiac arrest in more than one setting were classified by the location of the first arrest.
Comment 6:
Line 93: Please clarify whether the duration of IHCA is the same as the duration of IHCPR.
Response 6:
We revised Methods – Data definitions to define IHCA duration as the duration of active resuscitation (IHCPR); in our dataset these were operationally identical.
Comment 7:
Line 119: Present the inclusion and exclusion process more clearly, preferably as a flowchart.
Response 7:
In response to the reviewer’s suggestion, we added a study flow diagram that details the inclusion and exclusion criteria and derivation of the analytic cohort (Figure 2).
Comment 8:
Table 2: The “time to CPR” comparison shows means of 143.8 vs. 80.4 min with a p-value of 0.974. Given the large difference, this result is counterintuitive and likely reflects the highly skewed distribution (noted by the very large SD). Please confirm the statistical test used and consider presenting medians with interquartile ranges (IQR). Similarly, medians with IQR should be used for time to RTC, distance to RTC, distance to first hospital, and transfusion variables.
Response 8:
For skewed variables (e.g., time to CPR, time to RTC, distances, transfusion volumes), we re‑analyzed distributional assumptions (Shapiro–Wilk) and now present data as median (IQR) and compare groups using Mann–Whitney U tests. We updated Table 1&2 accordingly and added the sentence in Methods:
“Continuous variables are expressed as mean ± standard deviation for normally distributed variables and as median with interquartile range for non-normally distributed variables, and categorical variables as frequencies and percentages.”
Comment 9:
Please define how ROSC was determined. Does it refer specifically to ROSC achieved at your hospital after arrest?
Response 9:
We clarified in Methods – Data definitions:
“Return of spontaneous circulation (ROSC) was defined as the restoration of spontaneous circulation following the index traumatic cardiac arrest, irrespective of location (prehospital, referring hospital, or our regional trauma center).”
Comment 10:
Some variables (e.g., IHCA time, distance to first hospital) do not apply to the entire cohort but only to specific subgroups (in-hospital arrests or transferred patients, respectively). Presenting these variables without indicating the applicable sample size may be misleading. I recommend explicitly reporting the number of patients (N) for whom each variable is available, either within the table or in the table legend, to enhance clarity and transparency.
Response 10:
We revised Tables 2 and 3 to indicate N for variables applicable only to subgroups (e.g., IHCA time, distance to first hospital), either next to the variable name or in the table footnotes.
Comment 11:
Table 3: Some numbers and words (e.g., “Distance to first…”) are obscured and not readable. Please correct this.
Response 11:
We corrected formatting/word‑wrap issues so all labels (e.g., “Distance to first hospital”) and values are fully legible.
Comment 12:
Lines 155–157: Several hazard ratios are reported in the text, but it is unclear from which model (unadjusted, Model 1, 2, or 3) these estimates were derived. Since Table 3 provides different values across models, this should be specified. Alternatively, restrict the text to the fully adjusted model and direct readers to the table for details.
Response 12:
We revised Results 3.2 so that all HRs cited in the text are explicitly labeled as coming from the fully adjusted model (Model 3).
Comment 13:
Results 3.4 and Figure 2: In your interpretation of Figure 2, you highlight that transfer was associated with improved survival particularly in patients with oTCA, accident-to-CPR duration <30 min, and total CPR duration >30 min. However, a closer inspection of the forest plot suggests that the protective effect of transfer is more consistent across subgroups. Specifically, transfer appears favorable regardless of sex, age, duty time, or first hospital distance, with hazard ratios <1 in nearly all categories. Moreover, although the protective effect of transfer is attenuated when accident-to-CPR duration exceeds 30 min, it remains present. In addition, the subgroup with ISS ≥30 also shows a survival benefit with transfer, which is not clearly acknowledged in your discussion. I suggest revising the interpretation of Figure 2 to reflect the broader consistency of the transfer effect across subgroups, while also noting the relative differences in effect size.
Response 13:
We agree that our original wording over‑emphasized select subgroups. We revised Results 3.4 and the Discussion to state that the protective association of transfer is broadly consistent across most subgroups, with HRs < 1 in nearly all categories (sex, age, duty time, first‑hospital distance). We also note that while the effect is attenuated when accident‑to‑CPR exceeds 30 minutes, it remains protective, and that ISS ≥ 30 also shows a survival benefit with transfer. That Figure’s legend was updated to specify that estimates are from Model 3 and that HR < 1 favors transfer.
We appreciate the reviewer’s insightful comments, which substantially improved the rigor and clarity of our work. We believe the revisions address all concerns raised and enhance the manuscript’s methodological transparency, interpretability, and alignment with reporting best practices.
Thank you for your consideration.
Round 2
Reviewer 1 Report
Comments and Suggestions for Authors
Thank you for revising the manuscript.
Author Response
Comments 1: Thank you for revising the manuscript.
Response 1: We would like to thank you for reviewing our manuscript. The manuscript has been carefully rechecked and appropriate changes have been made.
Reviewer 2 Report
Comments and Suggestions for Authors
The authors have adequately addressed the concerns raised in the initial review, and the manuscript has improved considerably. However, the values in Table 3 are still not visible, and I would appreciate it if this error could be corrected prior to final publication.
Author Response
Comments 1: The authors have adequately addressed the concerns raised in the initial review, and the manuscript has improved considerably. However, the values in Table 3 are still not visible, and I would appreciate it if this error could be corrected prior to final publication.
Response 1: Thank you for your meaningful comment and suggestion. We have revised the size of Table 3 as requested. The manuscript has been carefully rechecked and appropriate changes have been made.